# Novel Dry Hyaluronic Acid–Vancomycin Complex Powder for Inhalation, Useful in Pulmonary Infections Associated with Cystic Fibrosis

**DOI:** 10.3390/pharmaceutics16040436

**Published:** 2024-03-22

**Authors:** María S. Magi, Yanina de Lafuente, Eride Quarta, María C. Palena, Perla del R. Ardiles, Paulina L. Páez, Fabio Sonvico, Francesca Buttini, Alvaro F. Jimenez-Kairuz

**Affiliations:** 1Departamento de Ciencias Farmacéuticas, Facultad de Ciencias Químicas, Universidad Nacional de Córdoba (UNC), Córdoba X5000GYA, Argentina; maria.sol.magi@unc.edu.ar (M.S.M.); yanina.de.lafuente@unc.edu.ar (Y.d.L.); celeste.palena@unc.edu.ar (M.C.P.); perla.ardiles@unc.edu.ar (P.d.R.A.); plpaez@unc.edu.ar (P.L.P.); 2Unidad de Investigación y Desarrollo en Tecnología Farmacéutica (UNITEFA), Consejo Nacional de Investigaciones Científicas y Técnicas (CONICET-UNC), Haya de la Torre y Medina Allende, Ciudad Universitaria, Córdoba X5000HUA, Argentina; 3Food and Drug Department, University of Parma, Parco Area delle Scienze 27/A, 43124 Parma, Italy; eride.quarta@unipr.it (E.Q.); fabio.sonvico@unipr.it (F.S.); francesca.buttini@unipr.it (F.B.)

**Keywords:** inhalable vancomycin, cystic fibrosis, polyelectrolyte–drug complex, hyaluronic acid, pulmonary infection

## Abstract

Polyelectrolyte–drug complexes are interesting alternatives to improve unfavorable drug properties. Vancomycin (VAN) is an antimicrobial used in the treatment of methicillin-resistant *Staphylococcus aureus* pulmonary infections in patients with cystic fibrosis. It is generally administered intravenously with a high incidence of adverse side effects, which could be reduced by intrapulmonary administration. Currently, there are no commercially available inhalable formulations containing VAN. Thus, the present work focuses on the preparation and characterization of an ionic complex between hyaluronic acid (HA) and VAN with potential use in inhalable formulations. A particulate–solid HA-VAN_25_ complex was obtained by spray drying from an aqueous dispersion. FTIR spectroscopy and thermal analysis confirmed the ionic interaction between HA and VAN, while an amorphous diffraction pattern was observed by X-ray. The powder density, geometric size and morphology showed the suitable aerosolization and aerodynamic performance of the powder, indicating its capability of reaching the deep lung. An in vitro extended-release profile of VAN from the complex was obtained, exceeding 24 h. Microbiological assays against methicillin-resistant and -sensitive reference strains of *Staphylococcus aureus* showed that VAN preserves its antibacterial efficacy. In conclusion, HA-VAN_25_ exhibited interesting properties for the development of inhalable formulations with potential efficacy and safety advantages over conventional treatment.

## 1. Introduction

Cystic fibrosis (CF) is a disorder that damages the lungs, gastrointestinal tract and other organs. It is a hereditary, potentially fatal disease, caused by a defective gene that affects the cells that produce mucus, sweat and gastrointestinal fluids due to defects in CF transmembrane conductance regulator protein [1,2]. Although it can affect different organs, pulmonary bacterial infections are the most frequent form of clinical presentation and the main cause of morbidity and mortality. In the lungs, viscous airway secretions result in inflammation and chronic infection which vary according to age; in fact, the progressive obstruction and long-term damage to the airways progressively leads to a decreased ability to clear secretions, causing increased rates of infections [3,4].

The most frequent respiratory infections associated with CF are caused *by Pseudomona aeruginosa* and *Staphylococcus aureus*, which are treated systemically with antimicrobials administered by different routes, in most cases intravenous or oral [5]. In particular, infection with methicillin-resistant *Staphylococcus aureus* (MRSA) is associated with significantly worse clinical outcomes. While intravenous vancomycin (VAN) is standard therapy for MRSA, in the respiratory setting, efficacy is reduced by relatively poor penetration into lung secretions and dose-limiting nephrotoxicity [6,7].

Inhaled antibiotics offer a pharmacotherapeutic alternative for the treatment of chronic respiratory tract infections; for example, they achieve high concentrations of drugs directly at the target site, mucosa of the lungs, while minimizing the risk of systemic side effects, improving both the efficacy and safety of the pharmacotherapy [8]. Nebulizer and dry powder formulations for inhalation are currently available for the treatment of infections caused by *Pseudomona aeruginosa*, containing tobramycin, aztreonam, levofloxacin or colistin. However, there are no commercially available inhaled antimicrobials that are effective against MRSA, i.e., ciprofloxacin and VAN formulations are in different stages of clinical development, denoting the high potential clinical relevance that local optimized treatments against MRSA could have to increase the survival of patients with CF [8,9]. In this line, the off-label use of inhaled VAN by nebulization of the intravenous formulation has been reported for the treatment of MRSA in the setting of CF, demonstrating a reduction in colonies but not eradication [10]. Nevertheless, inhaled formulations can present side effects, such as cough, bronchospasm, hoarseness and dysphagia, among others, which are frequently associated with the mucosal deposition of crystalline drug particles with slow dissolution rates [11].

Therefore, biocompatible drug delivery carriers are an interesting strategy to generate inhaled pharmaceutical products with improved properties beyond available therapies. Over the past decade, significant progress has been made in the development of new pharmaceutical technology platforms, based on polyelectrolyte-ionizable drug (PE-D) complexes, both in aqueous dispersions and solid material forms, with potential drug delivery applications [12,13,14]. The acid–base interaction between carboxylic groups of a PE and a basic group of a drug yields a high degree of counterionic condensation. PE-D complexes exhibit several unique and favorable properties to design smart or stimuli-responsive drug delivery systems. Based on this platform, modified drug release systems are designed with emphasis on approaching specific pharmacotherapeutic situations. These systems allow the efficacy, safety and reliability of the drug therapeutic action to be improved with recognized clinical impact [15]. In addition, they are an accessible and original alternative to developing sustained-release formulations and can be produced by a simple and easy-to-scale-up process [16].

In this context, hyaluronic acid (HA) is a natural, bioadhesive, and Generally Recognized As Safe (GRAS) polysaccharide. It has been reported to act as an oppositely charged drug delivery system [16]. Inhaled HA has the potential to protect against bronchoconstriction and hydrate the airway surface, resulting in an increase in airway fluid volume. Furthermore, hyaluronic acid (HA) is a vital constituent of the lung’s extracellular matrix, responsible for regulating fluid balance in the pulmonary interstitium.

Research has demonstrated that the properties of HA are primarily determined by its molecular size, with a distinct contrast between high-molecular-weight (HW) HA (>1 million Da) and small, low-molecular-weight fragments (150,000–300,000 Da) [17]. 

Although HW-HA fragments have anti-inflammatory and immunosuppressive properties, as well as anti-angiogenic and anti-inflammatory effects, the low-molecular-weight fragment is a potent pro-inflammatory molecule. 

Additionally, HW-HA has the ability to retain a significant amount of water in the extracellular matrix, producing viscous gels that could play an important role in both tissue homeostasis and biomechanical integrity [18].

Airway inflammation generates short fragments of HA through the degradation of structural HW-HA, which further promotes inflammation and airway hyper-responsiveness. However, HW-HA has been found to antagonize these effects and has been studied therapeutically in animal models of inflammatory airway diseases, such as CF. Additionally, HW-HA has recently been approved for use in upper airway disease and as an adjunct to hypertonic saline treatment in CF in some European countries. The addition of nebulized HW-HA to patients undergoing continuous hypertonic saline therapy results in an improved tolerability profile. This improves airway inflammation and hyper-responsiveness and effectively relieves symptoms such as cough, throat irritation and unpleasant taste associated with this treatment [19].

In this context, the aim of this work is to develop and carry out the physicochemical, pharmacotechnical and biopharmaceutical characterization of a particulate inhaled formulation based on an ionic complex between VAN and HA with optimized aerosolization behavior by the spray-drying technique which has antibacterial properties against MRSA of VAN.

## 2. Materials and Methods

### 2.1. Materials

Vancomycin hydrochloride (VAN) was kindly supplied by Lab. Química Luar SRL (Córdoba, Argentina). High-molecular-weight sodium hyaluronate (NaHA) was purchased from Pura Química^®^ (average MW 1.0–2.0 MDa, batch n° 313318, Córdoba, Argentina). Cation exchange resin (Amberlite^®^ IR-120 Fluka, Sigma-Aldrich, St. Louis, MO, USA); NaCl, KCl, NaHPO_4_ and KH2PO_4_ (Parafarm^®^, Buenos Aires, Argentina), NaOH and HCl 1N (analytical reagents, Anedra, Córdoba, Argentina), acetonitrile and acetic acid (VWR International, Fontenay-sous-Bois, France), sodium acetate and Span 80 (Merck KGaA, Darmstadt, Germany) and Müeller Hinton broth and agar (Britania, Buenos Aires, Argentina) were used as provided by suppliers. Ultrapure water (water purification NW-system, Heal Force Group, Shanghai, China) was used for all experiments.

### 2.2. Preparation of Hyaluronic Acid-Vancomycin Complex

As previously mentioned, the use of a high-molecular-weight NaHA is important for this formulation. In this regard, the viscosity of a 1% NaHA dispersion was determined at 60 rpm and a value of 121.9 ± 0.2 cPs was obtained, which is consistent with a high-molecular-weight NaHA [20].

Since it was not available commercially, the acidic form of HA was obtained from acid–base neutralization NaHA dispersions using a procedure reported by Battistini et al. [12]. Briefly, the purified HA dispersion (pH ≤ 2.5) was collected by eluting a 0.50% *w*/*v* NaHA dispersion through a column containing an anionic exchange resin, and then, the solid state was achieved by lyophilization (FreeZone 6, Labconco^®^, Kansas City, MO, USA, at −40 °C and 1 × 10^−3^ mPa of vacuum). 

Prior to use, determination of the ionizable carboxylic groups proportion per mass unit of HA (2.59 mmol/g) was performed by differential scanning potentiometry, according to the following methodology. An appropriate amount of HA powder (approximately 100 mg) was dissolved in ultrapure water containing 10 mL of HCl 0.05 N and then titrated with 20 mL of NaOH 0.05 N using an automatic acid–base titrator (Titrando^®^ 905, Software Tiamo^®^ 2.0 light, Methrom AG, Herisau, Switzerland). Another homolog titration containing an identical proportion of strong acid, but without adding HA, was used as reference. The results were in agreement with its monomeric structure and with the equivalents of a similar HA previously reported [12].

Series of complexes were prepared by neutralizing a 0.3% *w*/*v* aqueous dispersion of HA with an appropriate amount of VAN under constant stirring, referred to as HA-VANx, where x = 25, 50, 75 and 100%. The subscript ‘‘x’’ refers to the percentage of mole % of VAN which neutralizes carboxylic groups of HA. Then, the pH value of complex dispersion was adjusted, using NaOH 1N solution up to 7.0 ± 0.1.

All the complexes obtained were characterized with the aim of obtaining a physically stable dispersion without the sedimentation of aggregated particles or solid precipitates after adjusting the pH in order to minimize stability problems during spray drying.

The particulate complex was obtained by spray drying the aqueous dispersion with a mini Büchi-B290 Spray Dryer (Büchi^®^ Labortechnik, Flawil, Switzerland) using the following parameters: nozzle: 0.7 mm; inlet temperature: 120 °C; aspiration rate: 35 m^3^/h; feed rate: 4 mL·min^−1^; and air flow rate: 600 L/h. The outlet temperature was 70 °C. 

The powder obtained from the spray-drying process was removed from the collection vessel, dried under vacuum for 2 h and weighed on an analytical balance (Ohaus AR2140, Parsippany, NJ, USA, sensitivity 0.1 mg) to determine the yield. The manufactured powder was stored with a rubber stopper tightly sealed by a metal cup in a glass vial, protected from light and moisture and stored at 25 °C. The yield efficiency (*Y*) of the spray-drying process was calculated using the following Equation (1):(1)$$Y(\%)=\frac{Ws}{Wt}\times 100$$ where *Ws* is the weight of the powder obtained from the spray dryer and *Wt* is the total weight of the raw materials, HA and VAN. 

In parallel, a physical mixture (PM) of VAN and HA was prepared in a mortar with the same composition and proportion as HA-VAN_25_, which was used as a comparative control in the physicochemical characterization.

### 2.3. Physicochemical Characterization of HA-VAN_25_ in Solid State

The spray-dried powder of the HA-VAN_25_ complex was characterized at the molecular, particulate and bulk level using different analytical and physical methods in order to evaluate the suitability of HA-VAN_25_ as a potential inhalation powder.

#### 2.3.1. Infrared Spectroscopy 

The FTIR spectra of the complex, HA-VAN_25_, PM and the raw materials were collected in the range from 4000 to 650 cm^−1^ using an FTIR spectrometer (Cary 630, Agilent^®^, Santa Clara, CA, USA) set to a resolution of 4 cm^−1^ and were scanned 40 times per spectrum, equipped with the specific analysis software, OMNIC 8.3 spectrum software (Thermo Scientific, Waltham, MA, USA).

#### 2.3.2. Thermal Analysis 

Differential scanning calorimetry (DSC) and thermogravimetric analysis (TGA), using a TA^®^ Discovery series instrument (TA Instruments, New Castle, DE, USA), were used to evaluate the thermal behavior of the complex, the PM and the raw materials. The results were processed and analyzed using TRIOS^®^ software v4.1.1 (TA Instruments). Approximately 2 mg of each sample was weighed and sealed in non-hermetic aluminum pans. DSC analysis was performed in a temperature range starting from room temperature (25 °C) until sample decomposition, between 190 and 215 °C, using a heating ramp of 10 °C/min under N^2^ atmosphere (50 mL/min). TGA analysis was carried out under the same conditions as DSC analysis but with a temperature range from room temperature to 400 °C.

#### 2.3.3. X-ray Diffraction

Powder X-ray diffraction (PXRD) patterns were obtained to evaluate the crystalline or amorphous state of VAN, HA and HA-VAN_25_. For this purpose, an X-ray powder diffractometer (PW1800, Philips, Amsterdam, The Netherlands) using Cu Ka radiation (λ = 1.5418 Å, tube at 40 kV, 100 mA) was used. Data were collected over an angular range of 5 to 60° 2θ/θ using a step method, with a step of 0.02 and scan rate of 1 seg per step.

### 2.4. Powder Characterization 

#### 2.4.1. Scanning Electron Microscopy (SEM)

Images were captured using an electron microscope (FE-SEM Sigma, ZEISS Instrument, Oberkochen, Germany) with magnification of 1000× for VAN and 1000× but also 10,000× for HA-VAN_25_. The samples were prepared by dispersing 1–2 mg of powder directly on the carbon tape, placed on the aluminum stubs. The surface morphology and texture of the particles were investigated in plan-view by using a 1 kV electron beam acceleration voltage. 

#### 2.4.2. Density Determination and Flow Properties

The flow properties of pharmaceutical powders are frequently evaluated from the bulk and tap densities determinations, in accordance with United States Pharmacopeia (USP) specifications [21]. To determine bulk density (*δ_B_*), an exactly weighed amount of HA-VAN_25_, between 0.5 and 1 g, was gently introduced into a 10 mL calibrated measuring cylinder, avoiding compaction. The powder was carefully leveled, and the volume was read to the nearest graduated unit. To measure tap density (*δ_T_*), the powder contained in the cylinder was manually tapped until no further changes in the volume were observed. From these data, Carr’s index and the Hausner ratio were calculated according to Equations (2) and (3), respectively [22]: (2)$$\mathrm{Carr}’s\;\mathrm{index}\;(\%)=100\;\times \;\frac{\delta T-\delta B}{\delta T}$$
(3)$$\mathrm{Hausner}\;\mathrm{ratio}=\frac{\delta T}{\delta B}$$
where *δ_T_* is the tap density and *δ_B_* is the bulk density. Each sample was tested in triplicate. 

#### 2.4.3. Particle Size Distribution by Laser Diffraction

The measurement of particle size distribution (PSD) of dry powder HA-VAN_25_ was performed using a Mastersizer 3000 (Malvern Instruments Ltd., Worcestershire, UK) laser diffractor. All the samples were dispersed using a dry powder feeder (Aero S, Malvern Panalytical, Malvern, UK) at a dispersive air pressure of 4.0 bar and 3 mm sample flow control, and the vibration feed rate was set to 45%. The results of the median volume diameter of the powder were expressed as *Dv*90, *Dv*50 and *Dv*10, namely, cumulative undersize volume diameters at 90%, 50% and 10% of the particle population, respectively. Another parameter measured was the *SPAN* value, which is the distribution width, and was calculated using the following Equation (4):(4)$$\mathrm{SPAN}=\frac{Dv90-Dv10}{Dv50}$$

### 2.5. In Vitro Biopharmaceutical Performance of HA-VAN_25_ Complex

#### 2.5.1. Aerodynamic Performance Assessment

The aerodynamic distribution of the powders was assessed using the next-generation impactor (NGI) at a flow rate of 65 L/min (Copley Scientific Limited, Nottingham, UK). The 30 mg of aerosolized powder which contained 10 mg of VAN, according to the formulation, was loaded in an HPMC QUALI-V I size 3 capsule (Qualicaps, Madrid, Spain) and inserted in an RS01^®^ inhaler device (Plastiape, Lecco, Italy). After aerosolization, the powder deposited in the different NGI parts was collected using distilled water and the VAN was quantified by HPLC (see next section).

Different parameters were calculated according to USP specifications for dry powder inhalers [23]. Between these, the emitted dose (ED) corresponds to the amount of VAN that leaves the inhaler device and enters the NGI; the emitted fraction (EF) is the ratio between the ED and the total mass of powder in the inhaler device. From the amount of powder deposited on the impactor and its ratio to ED, the mass median aerodynamic diameter (MMAD, dae, where 50% of the population is smaller than that value), the fine particle fraction (FPF, fraction of powder with dae < 5 μm) and the extra-fine particle fraction (EFPF, fraction of powder with dae < 3 μm) were calculated.

#### 2.5.2. Analytical Quantification of Vancomycin

HPLC was performed to determine the VAN content in the spray-dried complex using the Agilent equipment 1200 LC series (Agilent Technologies, Santa Clara, CA, USA) equipped with a UV-Vis detector, and the data were analyzed by OpenLab CVS Chem Station software (rev.c.01.06 v.A.04.02, Agilent Technologies). A reverse-phase and isocratic method was employed, along with a 150 × 3 mm Luna C18 3 μm (Phenomenex, Torrance, CA, USA) column at 30 °C. VAN was eluted using a flow rate of 0.45 mL/min and an injection volume of 20 μL with a mobile phase formulated with acetonitrile and acetate buffer (pH 3.50 8:92 *v*/*v*). The detector was set at 240 nm, and the retention time was around 1.7 min. The analytical method was validated for linearity, precision and accuracy in the appropriate concentration range of analysis in accordance with ICH Guideline Q2. Calibration curves were constructed using five concentrations of VAN dissolved in deionized water in a range from 12.5 to 1000 µg/mL, in triplicate. Each standard was injected five times, and the linearity and precision expressed as the relative standard deviation (RSD%) of the assay was calculated. The correlation coefficient (R^2^) for the calibration curve was 0.9996 with a linearity range of 12.5–1000 µg/mL. The reproducibility was considered acceptable as the RSD% values obtained were, for all the concentrations tested, <2% (limit defined by the ICH guidelines). The LOQ (lower limit of quantification) was 0.25 μg/mL (RSD% 1.72), and the LOD (limit of detection) was 0.125 μg/mL (RSD% 1.22). 

#### 2.5.3. Dissolution Study

Dissolution tests of VAN and HA-VAN_25_, were carried out following USP Method I (SOTAX AT 7 Smart, Westborough, MA, USA) using 500 mL of phosphate-buffered solution (PBS) at pH = 7.4 and 37.0 ± 0.5 °C as dissolution medium, with a basket at 100 rpm speed. 

Dissolution tests of new formulations are usually compared to a reference. However, there are no commercial formulations of VAN for the inhalation route. For this reason, an amount of powder containing 150 mg of VAN as raw material and the equivalent of VAN in the HA-VAN_25_ complex were tested. The latest formulation was also sieved using a 560–630 μm sieve prior to the test. Each sample was incorporated into the dissolution medium and 0 and 2 mL aliquots were taken at predetermined times (5, 10, 15, 30, 45, 60, 90 and 120 min); these extracted volumes were replaced with a fresh thermostated medium. The aliquots were filtered through a cellulose filter before dilution. The concentration of dissolved VAN was determined by UV-Vis spectrophotometry (Evolution 300, Thermo Electron Corporation, Waltham, MA, USA) at 280 nm. A calibration curve, in triplicate, was designed with six different concentrations of VAN between 55 and 280 µg/mL dissolved in deionized water, obtaining an R^2^ of 0.9985. 

The results were expressed as the mean of three determinations with their SD. 

#### 2.5.4. Franz’ Cell Diffusion Study

VAN release from complex dispersions was performed at 37.0 ± 0.1 °C in bicompartmental Franz cells, and the donor and the receptor compartment were separated by a semisynthetic cellulose membrane (MW cut-off 12 kDa, Sigma-Aldrich, St. Louis, MO, USA). In the donor compartment of each cell, 1 mL containing 11 mg of VAN or HA-VAN_25_ in the same proportion of VAN was introduced. Sufficient water or PBS (pH = 7.4), approximately 16 mL, were used as the receptor medium. Samples of 0.8 mL were taken at predetermined time intervals, and the extracted volume was replaced with fresh medium heated to 37 °C. The concentration of released VAN was analyzed by UV-Vis spectrophotometry at 280 nm. Assays were carried out in triplicate, and immersion (sink) conditions were maintained.

The profiles obtained were statistically compared using the difference factor, *f*_1_, presented in Equation (6). 

In addition, the release data were processed using the empirical power equation proposed by Peppas [24] to evaluate the kinetics and mechanism of drug release as follows:(5)$$\frac{M_{t}}{M_{\infty }}=k\times t^{n}$$
where *M_t_* is the amount of drug permeated at time *t*; *M*_∞_ is the initial amount of drug in the donor compartment; *k* is the kinetic constant; and *n* is the diffusion exponent characterizing the release mechanism, *n* = 0.5 (Fickian diffusion), 0.5 < *n* < 1 (non-Fickian diffusion) and *n* = 1 (cero order, controlled release). Equation (5) is valid in the range up to 60% of the released drug. 

### 2.6. Antibacterial Activity Assays

The samples tested were HA, VAN and HA-VAN_25_. Firstly, an agar diffusion assay was performed to determine the sensitivity on *S. aureus* strains, both methicillin-sensitive (MSSA) and -resistant (MRSA). For this purpose, a plate containing Müeller Hinton Agar (MHA) was inoculated with three ATCC strains, 29213, 25923 and 43300 of *Staphylococcus aureus*, where the last one corresponds to the MRSA strain. The strains used belong to the strain collection of the microbiology laboratory (Department of Pharmaceutical Sciences, the Faculty of Chemical Sciences, UNC). The stock culture was stored at −80 °C in trypticase soy broth (TSB, Britania) with 10% (*v*/*v*) glycerol. Then, wells were made where 40 μL of the samples were incubated at 37 °C for 24 h. The inhibition halo formed by each sample was measured. Secondly, the minimum inhibitory concentration (MIC) and minimum bactericidal concentration (MBC) against MSSA and MRSA strains were determined using the microplate dilution method according to CLSI standards [25]. From the stock solutions (SSs) of VAN, HA or HA-VAN_25_ serial dilutions (factor 2) were performed in Müeller Hinton Broth (MHB) on a multiwell plate. Samples of 100 μL of standardized bacterial inoculum were added to each well and incubated at 37 °C for 24 h. The MIC corresponds to the lowest concentration that inhibits microbial growth and was determined by verifying the absence of turbidity with the naked eye. After the MIC readings, dilutions without bacterial growth were plated on MHA and incubated for 24 h at 37 °C to determine the MBC. The MBC value was considered to be the lowest sample dilution in which no microbial growth was observed, and therefore, 99.9% of the initial inoculum had been reduced.

### 2.7. Statistical Analysis

The data are expressed as mean ± standard deviation (SD). The statistical analysis of the agar diffusion assay was performed using Student’s *t*-test. *p* < 0.05 was accepted as the level of statistical significance. InfoStat^®^ statistical software (2020 v., https://www.infostat.com.ar/, accessed on 8 March 2024, Córdoba, Argentina) was used.

The dissolution profiles obtained were statistically compared using the difference factor (*f*_1_, Equation (6)), with two profiles being considered different when the value of *f*_1_ calculated between them was greater than 15.
(6)$$f_{1}=\frac{\sum_{i=1}^{k}\left|R_{i}-T_{i}\right|}{\sum_{i=1}^{k}R_{i}}\;\times \;100$$

## 3. Results and Discussion

### 3.1. HA-VAN_25_ Complex Preparation 

The PE-D complex was successfully obtained using VAN and HA. It was important to use HA instead of sodium salt to favor the interaction with VAN, since it also has lower viscosity than NaHA, which reduces the amount of complex that sticks to the spray dryer walls, increasing the process yield. The inclusion of HA in this formulation might play a key role in airway diseases with a high inflammatory component, such as CF [19]. 

All HA-VANx complexes, where x represents 25, 50, 75 and 100%, were obtained and characterized. The HA-VAN_25_ complex, which was formed spontaneously after dispersing the raw materials under constant stirring, was selected as the most appropriate complex composition that remains as a transparent colloidal dispersion after pH correction and prevents the formation of agglomerates and solid sedimentation. This complex has a mass to mass % ratio of HA:VAN of 52:48. The addition of NaOH in order to obtain a more biocompatible pH value did not produce any changes in the appearance of the formulation, like darkening or precipitation, so it could be inferred that the system is physically stable. This dispersion was successfully spray dried, obtaining a yield of 69% *w*/*w* of the microparticulate HA-VAN_25_ complex in the solid state. 

### 3.2. Physicochemical Characterization of HA-VAN_25_ in the Solid State

The solid-state integral characterization of HA-VAN_25_ microparticles was carried out using several characterization techniques including FTIR analysis, PXRD, TGA and DSC.

Figure 1 shows the comparative FT-IR spectra, where the bands that could serve as indicators of acid–base interactions were identified in order to analyze the changes attributable to the ionic association between HA and VAN. Regarding the FT-IR spectrums of the raw materials, HA showed the characteristic bands related to carboxylic groups, such as those at 3357 cm^−1^ corresponding to the O-H tensile vibration of the COOH group, at 1732 cm^−1^ attributable to the C=O carbonyl stretching vibration of the COOH group and at 1314 cm^−1^ corresponding to the C-O bond of the carboxyl group of COOH [12,26]. In the VAN spectrum, a broad band at 3280 cm^−1^ corresponding to the overlapping O-H and N-H tensile vibrations of the acid and amino groups was observed. In addition, the C=O stretching vibration at 1647 cm^−1^ overlapped with the N-H bending vibration of the amino groups present, and the signal at 1225 cm^−1^ was attributed to the C-O bond stretching [27]. 

Comparing the HA-VAN_25_ complex with the raw materials and its PM, notable differences were observed. The band at 1732 cm^−1^, corresponding to the C=O of the COOH group, evident in both HA and the PM, was absent in the complex. Furthermore, a band emerged at 1401 cm^−1^, attributed to the stretching of the C-O bond in the carboxylate group (COO-), which was not present in the PM. These findings suggest a potential ionic interaction between HA and VAN, as only the bands corresponding to the ionized groups were evidenced.

The DSC and TGA thermograms, depicted in Figure 2, illustrate the thermal characteristics of the raw materials (HA and VAN), the HA-VAN_25_ complex and the PM. The TGA analysis reveals two significant weight loss events in VAN. The initial extensive event, starting from room temperature (25 °C) to 100 °C, is attributed to material dehydration resulting from the loss of water adsorbed on the solid particle surfaces, with a weight percent loss of (9.8 ± 0.7)%. This event aligns with the DSC thermogram, represented by a broad and nonspecific endotherm within the same temperature range. The second TGA event likely corresponds to the decomposition temperature, initiating at 208 °C, as evidenced by the sustained weight loss in the TGA curve. Additionally, it is noteworthy that the melting temperature of pure VAN could not be observed under the assay conditions, conducted up to 215 °C. This is consistent with the findings of other authors who investigated the thermal properties of VAN [28].

On the other hand, the DSC thermogram of HA exhibits an initial endothermic peak followed by an exothermic peak at approximately 180 °C, coinciding with mass loss in TGA, indicative of the initiation of its decomposition. The glass transition of pure HA could not be observed at the test temperatures. This aligns with prior findings, as reported by other authors, who similarly demonstrated a multi-stage decomposition process for HA where the initial degradation step involves dehydration from the compound structure, followed by a second stage near 200 °C, consistent with our results, and two additional peaks observed at 320 °C and 415 °C [28]. 

The thermograms of the HA-VAN_25_ complex exhibited weight loss in TGA and an endothermic dehydration peak in DSC between room temperature and 100 °C, mirroring the behavior of the precursors. However, unlike the PM, which decomposes at the same temperature as HA (the precursor that decomposes the first), the TGA spectra of the HA-VAN_25_ complex revealed a shift in the decomposition temperature towards higher temperatures compared to HA. Consequently, the formation of a complex with VAN appeared to shift HA decomposition at temperatures higher than 180 °C, thereby enhancing the polymer’s stability. This observation suggests the presence of intermolecular forces between HA and VAN, influencing their thermal behavior, which aligns with the previously presented FT-IR results.

The p-XRD patterns of VAN, HA and HA-VAN_25_ were determined. In Figure 3, it can be observed that both precursors, the PM and the complex are amorphous, evidenced by the absence of diffraction peaks, a consistent observation reported by previous researchers and in alignment with the previously mentioned DSC spectra [29].

It has been reported that binary PE-D complexes, obtained by mixing a PE with a drug of opposite charge in a convenient medium capable of dissolving one or both components, result in amorphous solid materials where the drug is ionically bound to the polymeric carrier [30]. 

Generally, amorphous powders are favored over crystalline counterparts for pulmonary drug delivery due to the numerous advantages they offer [31]. For example, the deposition of crystalline particles can induce an inflammatory response, as reported by other authors who studied the inhalation of crystalline rifapentine particles [32]. Currently, an inhaled formulation containing tobramycin (TOBI^®^, podhaler^®^, Novartis AG, Buenos Aires, Argentina) is commercially available for the treatment of *Pseudomona aeruginosa* infections associated with CF, which has an amorphous structure similar to that observed in our HA-VAN_25_ system [33].

Despite the p-XRD patterns obtained, revealing amorphous structures in both HA-VAN_25_ and its precursors, HA and VAN, the comprehensive information provided by FT-IR, DSC and TGA suggests an ionic interaction between the carboxylic groups of HA and the basic groups of VAN.

### 3.3. Powder Characterization

There are different physical factors that can affect the aerosolization and therefore the deposition in the deep lung of dry powders, such as particle size distribution, the flowability of the formulation, as well as particle density and shape [34,35,36].

As is known, particles having a geometric size range between 1 and 5 µm are, in principle, suitable to reach the deep lung since particles with sizes higher than 5 µm can be deposited in the upper airways, whereas particles smaller than 1 µm can get exhaled [34]. 

SEM images were taken after the spray-drying process of the complex as well as the VAN as the raw material at different magnification values. According to the micrographs observed in Figure 4A,B (magnification 1000×), VAN showed a flat and lamellar structure with the irregular shape and size of the particles between 20 and 40 µm, which is considered too large for inhalation purposes. On the contrary, when the HA-VAN_25_ complex was observed, much smaller particles were created (Figure 4B), and in the image with a magnification 10 times higher (Figure 4C, 10,000×), it was noticed that most of the particles presented a size between 3 and 4 µm. Regarding the morphology of HA-VAN_25_, the particles showed a characteristic hemispherical hollow shape with smooth surface and edges, as was previously described by Martinelli et al. [37] and Ceschan et al. [38] for similar hyaluronate sodium salt-based spray-dried particles. The formation of agglomerates was noticeable, as it was also evident macroscopically, and it could also be seen structures of one-inside-other particles in some cases.

The use of spray drying as a process for the obtention of the solid can be beneficial in the morphology of the particles, as already demonstrated by the spray-dried inhalation powder of colistimethate sodium for lung infections in CF [39]. Advantageous properties can be associated with the use of this technique such as the shape and diameter uniformity of the particles, which can provide good flowability, while the formation of hollow particles allows low-density structures, reducing the aerodynamic diameter and improving the aerosolization performances [40,41].

In this context, the determination of the bulk density and flowability properties of inhalable dry powders provides important material quality attributes directly related to the aerosolization behavior [42,43]. Taking all of this into consideration, the bulk and tapped density of the HA-VAN_25_ complex were measured showing a value of (0.13 ± 0.02) g/mL and (0.18 ± 0.03) g/mL, respectively. These values are reasonable for inhalable powders since tapped densities below 0.4 g/mL have been reported in many works as cut-offs for determining good aerodynamic characteristics [44].

However, the smaller the sizes of particles are, the higher the particle–particle interaction is, which is related to Van der Waals forces concerning high-contact areas; therefore, these cohesive forces can lead to poor flow properties and the formation of agglomerates [42]. Particle agglomerates and poor flowability can impact the aerosolization performance of the powder formulation that ultimately can remain in the inhaler after patient inhalation, resulting in low emitted doses [36]. On the other hand, it is well established that aerosolization performance in terms of the fine particle dose of the emitted powder is optimal with smaller particle sizes; therefore, the particle size and density of inhaled powders has to be engineered to balance their properties in order to provide the aerosolization performance as well as proper flowability. 

In order to study the flow properties of the HA-VAN_25_ complex, Carr’s index (CI) and the Hausner ratio (HR) were used according to Equations (2) and (3), obtaining values of 26.9 ± 5.8 and 1.4 ± 0.1, respectively. The high CI and HR values indicate a powder with poor flow characteristics based on the scale of powder flowability according to USP [22]. 

With regard to flowability, the less compressible the powder is, the less cohesive it will be, and for that reason, the better flow it will have [44,45]. In the case of HA-VAN_25_, the differences between bulk and tap densities evidenced high compressibility, leading to a poor flow property, which has also been seen previously for sodium hyaluronate spray-dried particles by Martinelli et al. [37].

In addition, in a preliminary determination of the capability of the powder to reach the deep lung, the particle size distribution was measured by laser diffraction, which is correlated with the aerodynamic particle size. HA-VAN_25_ presented a Dv90 = (6.37 ± 0.07) µm, Dv50 = (2.90 ± 0.02) µm and Dv10 = (1.19 ± 0.02) µm, meaning that 90%, 50% and 10% of the particle population, respectively, has a particle size below those respective values. In addition, a SPAN value of (1.79 ± 0.03) was measured, which indicates a monodisperse distribution of particle size. These are promising results, where more than half of the population has a geometric size below 5 µm within the acceptable range being suitable for deep lung deposition using only the HA-VAN_25_ complex, without other pharmaceutical excipients. 

### 3.4. In Vitro Biopharmaceutical Performance of HA-VAN_25_ Complex

An in vitro deposition distribution study after the aerosolization of complex powder loaded in rigid HPMC capsules using the NGI equipment was performed. Table 1 summarizes the parameters measured, including ED, EF, FPF, Extra-FPF and MMAD, which can be related to the capability of the powder to be transported through the patients’ airways. Despite the poor flowability evidenced, the powder provided a good emitted fraction (above 80%) when aerosolized with the RS01 device. It appears evident that the agglomerates of HA-VAN_25_ spray-dried microparticles highlighted by SEM analysis and the cohesive forces providing poor flow were disrupted during the capsule spinning, powder extraction and subsequent impaction on the walls and de-agglomeration grid of the RS01 device. The FPF value showed that about 43% of the powder has the potential to reach the deep lung since the particles have an aerodynamic diameter lower than 5 µm, while 26% of the particles can go deeper until the alveolus. The MMAD showed that 50% of the population has a size lower than 4.29 µm, which is in good agreement with the PSD obtained by laser diffraction.

The drug’s aerodynamic particle size distribution reported in Figure 5 revealed that approximately 16% of the VAN was not emitted from the capsule and the inhaler device, while 22% was retained in the IP, which simulates the throat. Those percentages, besides the amount of VAN deposited in stage 1 (9%), correspond to the quantity of drug that will not reach the deep lung. The presence of a high amount of drug from stage 2 onwards is considered desirable for lung deposition [46]; since this stage has a cut-off diameter of 4.46 µm at the flow rate of 65 L/min, almost half of particles of HA-VAN_25_ are in these stages. The higher values of VAN in stage 2 and 3 in comparison with the following is consistent with the MMAD value obtained. 

Despite the absence of vehicle excipients frequently used in dry powder inhalable formulations, such as lactose or mannitol, the HA-VAN_25_ powder presented satisfactory efficiency for pulmonary administration by itself. Moreover, as demonstrated by the optimal value of the VAN emitted fraction, the formation of weak agglomerates could result in the easy disaggregation either for the inspiratory force generated by the patients and/or the collisions between the particles or particle walls inside the inhaler device, leading to a proper EF [47]. The suitable aerosolization performance observed is consistent with previous results where the powder showed a low bulk density; the morphology presented a hollow hemispherical shape and the Dv50 value was about 2.9 µm; all of these results together influence the aerodynamic performance which would lead to adequate pulmonary deposition [48]. 

In a previous study by Sullivan et al. (2015) [28], the in vitro and in vivo performance of dry powder vancomycin hydrochloride (VAN) without further processing was investigated in intubated rabbits, revealing promising results. Intubated rabbits administered a 1 mg/kg dose of VAN via inhalation demonstrated a comparable AUC to those receiving the same dose through a single bolus IV infusion. Notably, inhaled VAN exhibited reduced C_max_ and increased T_max_, indicating a more sustained pulmonary level of the drug. However, the physicochemical and flow properties obtained by these authors were not the most suitable for inhalation therapy; for example, the VAN showed different particle sizes and shapes and presented high bulk and tap densities of (0.35 ± 0.01) g/cm^3^ and (0.51 ± 0.01) g/cm^3^, respectively. The FPF value was no more than 26%, and the MMAD value was close to 7 μm, accompanied by a substantial geometric standard deviation [28]. For this reason, our dry powder formulation, based on HA-VAN_25_ with better physicochemical and flow properties as mentioned before, is expected to overcome these results in terms of in vivo VAN performance, emphasizing both its efficacy and safety.

Following the impaction process, dry powders deposited within the lungs are required to undergo complex drug absorption processes that include wetting, dissolution and diffusion. Since no established specific dissolution method for inhalation products is defined by pharmacopoeias and guidelines, the traditional basket apparatus dissolutor (USP Apparatus 1) was used, which has a relatively large volume of dissolution medium where the powder is dispersed. The dissolution profiles of VAN and the HA-VAN_25_ complex are depicted in Figure 6. It was evident that the complete amount of pure VAN dissolved within 5 min, while the dissolved proportion of VAN from HA-VAN_25_ was less than 40% at the same time. The comparison between the VAN and HA-VAN_25_ dissolution profiles showed *f*_1_ = 40.6, denoting significant differences (*f*_1_ > 15). Given the hydrophilic nature of VAN, prompt drug dissolution was anticipated. Conversely, the formation of an ionic complex with HA leads to a slower dissolution rate. Additionally, it is noteworthy that the complex, when in contact with the dissolution medium, undergoes gelation and slight swelling due to the presence of the polymer, potentially contributing to the observed reduction in the dissolution rate.

Beyond the differences in the dissolution profiles between VAN and HA-VAN_25_, it is noticeable that more than 85% of the loaded VAN dissolves within the initial 30 min. This aspect is crucial, as inadequate drug dissolution could potentially induce lung irritation, cause local side effects and trigger processes like macrophage phagocytosis or the mucociliary clearance of solid particles, ultimately leading to a rapid reduction in lung dose [49].

In addition, bicompartmental Franz cells were used in order to study the release behavior of the aqueous HA-VAN_25_ complex dispersion in comparison with a solution of pure VAN in two different media: water and PBS pH 7.4. 

It can be seen in Figure 7, when water was used as a receptor medium, a fast diffusion of VAN across the dialysis membrane from its solution was observed. However, the release rate of VAN from the aqueous dispersion of its complex was substantially slower. This response can be attributed to the reservoir behavior usually described in PE-D complexes, where the dissociation of the ionic complex between HA and VAN is the determining step in the VAN release toward the semipermeable membrane [12,50]. In addition, the electrostatic attraction between the oppositely charged macroions present in the donor compartment makes the diffusion of ionic species more restricted [14].

When water was replaced by PBS pH 7.4 solution (as a simulated physiological receptor medium), a significant increase in VAN diffusion from HA-VAN_25_, in comparison with its release profile in water, was observed, reflected by an *f*_1_ value of 72.6. It is known that the presence of dissolved ions in the receptor medium (from PBS solution) promotes ion exchange from the macromolecular complex microenvironment, increasing the proportion of free ionic species of VAN and, consequently, a significant increase in drug release rates [12].

On the other hand, the similarity of VAN release profiles from pure VAN solution and from the complex, both in PBS medium, (*f*_1_ = 15), could be related with the diffusion of ions from the receptor compartment to the donor compartment, promoting the dissociation of VAN HCl into VAN and HCl separately, shifting the acid–base equilibrium towards the non-ionized VAN (VAN base), increasing their proportion in solution. The VAN base presents a lower solubility (0.225 mg/mL) than VAN HCl in water (>100 mg/mL), i.e., the solubility decreases over 400 times [51]. This could be the reason why VAN diffuses faster in water than PBS 7.4, and, at the same time, with a similar release profile than the complex in the saline solution. However, additional experiment assays would be necessary to confirm this observation. 

Finally, the kinetic analysis of the release profiles showed a strong fit to the power equations (R^2^ 0.96 and 0.99) above the Higuchi diffusion model. The values of diffusional exponent *n* between 0.5 and 1 suggest an anomalous or non-Fickian release behavior (Table 2). Consequently, the dissociation of ionic pairs between HA and VAN and the later diffusion of VAN from HA-VAN_25_ seem to be the principal control mechanisms of drug delivery. The polymer chains reorganize slowly, whereas the diffusion process proceeds very rapidly and leads to anomalous time-dependent effects, as was observed in the majority of PE-D systems [52]. 

### 3.5. Antibacterial Activity Tests 

The agar diffusion test was carried out to assess the susceptibility against *Staphylococcus aureus* reference strains, specifically strains ATCC n° 29213, 25923 and 43300. The results, presented in Table 3, indicate that HA does not exhibit inhibition, displaying a halo of 8 mm corresponding to the diameter of the original well. HA itself is not considered inherently antimicrobial. In contrast, VAN demonstrates an inhibition halo ranging between 29 and 31 mm, depending on the strain, and the complex exhibits a similar inhibition pattern. Notably, none of these samples demonstrate reduced inhibition against the MRSA strain.

The antibacterial efficacy of HA-VAN_25_ was assessed by examining MIC and MBC against *Staphylococcus aureus*. Table 4 reveals that not only does VAN exhibit antibacterial activity, but so does HA-VAN_25_ against MRSA and MSSA. Remarkably, HA alone does not demonstrate inhibitory or bactericidal activity in concordance with the absence of halo inhibition, as previously mentioned. In contrast, VAN exhibits the same MIC and MBC values, consistent with its recognized bactericidal nature, and these values are between 4 and 8 µg/mL, in agreement with previous findings [53,54].

Notably, HA-VAN_25_ demonstrates comparable antimicrobial activity with VAN, underscoring that the formation of the complex preserves the well-established antibacterial efficacy of VAN against both MSSA and MRSA strains while simultaneously offering the advantages of being complexed with HA. 

Further pharmacokinetics and security studies will be needed in order to evaluate the in vivo performance of inhaled HA-VAN_25_ powder compared with intravenous VAN in animal experimental models.

## 4. Conclusions

A dry powder based on the biocompatible HA-VAN_25_ complex was successfully obtained using a simple and scale-up method without organic solvents. The characterization of the solid state confirmed the ionic interaction between HA and VAN and the formation of a new entity with properties different to the raw materials. The studies of the morphology, size and bulk and tap densities of the powder revealed that it is adequate to reach deep lungs with a spherical shape, decreasing the possibility of irritating the lungs. The HA-VAN_25_ complex aerodynamic performance showed that it is suitable for pulmonary administration. The dissolution of the complex was fast, and the in vitro release showed an extended release of VAN. In addition, the HA-VAN_25_ complex showed the preservation of the antibacterial efficacy of VAN while simultaneously offering the advantages of being complexed with HA. This unique combination not only maintains the antimicrobial activity of VAN but also provides additional benefits, such as mitigating irritation and inflammation and hydrating the airways. These characteristics make the complex highly promising for potential applications in respiratory health, promising a more effective and well-tolerated approach to combat bacterial infections caused by MRSA in patients with CF, mainly improving their quality of life. 

## Figures and Tables

**Figure 1 pharmaceutics-16-00436-f001:**
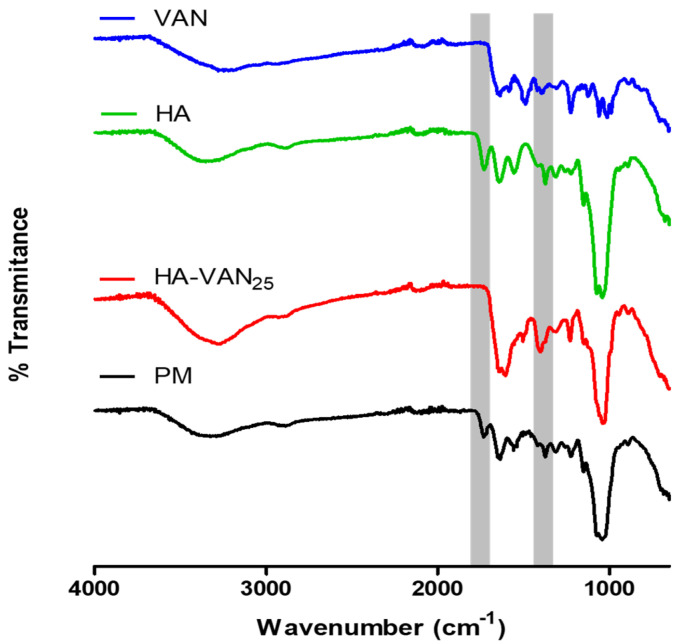
Overlapped Fourier transform infrared spectroscopy spectra of HA-VAN_25_ complex; raw materials, vancomycin (VAN) and hyaluronic acid (HA) and the physical mixture (PM).

**Figure 2 pharmaceutics-16-00436-f002:**
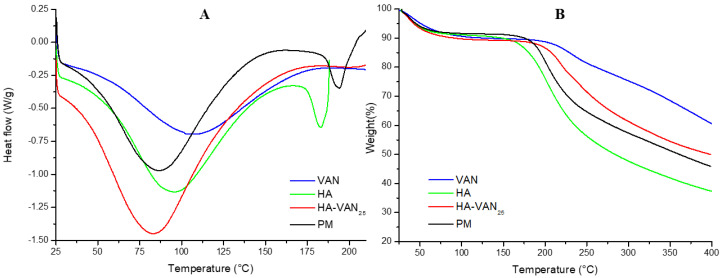
Thermograms of differential scanning calorimetry (**A**) and thermogravimetric analysis (**B**) of the HA-VAN_25_ complex, the raw materials (VAN and HA) and the PM.

**Figure 3 pharmaceutics-16-00436-f003:**
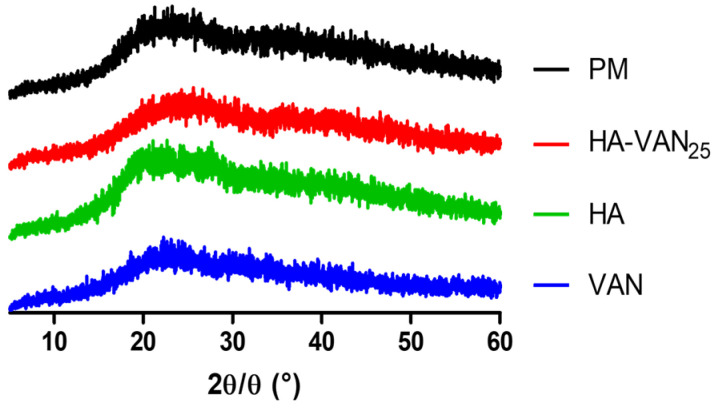
Powder X-ray diffraction patterns of HA-VAN_25_, the raw materials (VAN and HA) and the PM.

**Figure 4 pharmaceutics-16-00436-f004:**
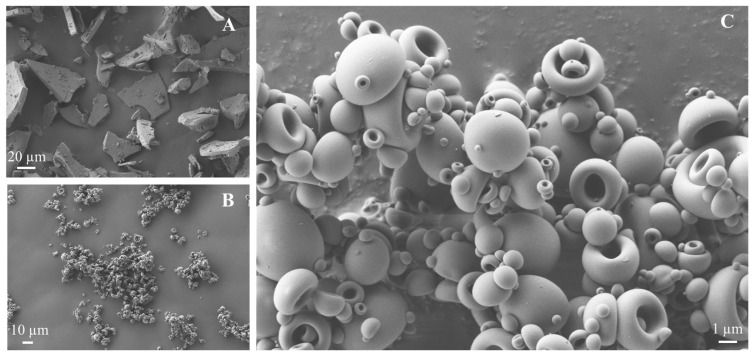
Scanning electron microscopy micrographs of VAN (**A**), HA-VAN_25_ (**B**) with the same magnification (1000×) and HA-VAN_25_ (**C**) with a higher magnification (10,000×).

**Figure 5 pharmaceutics-16-00436-f005:**
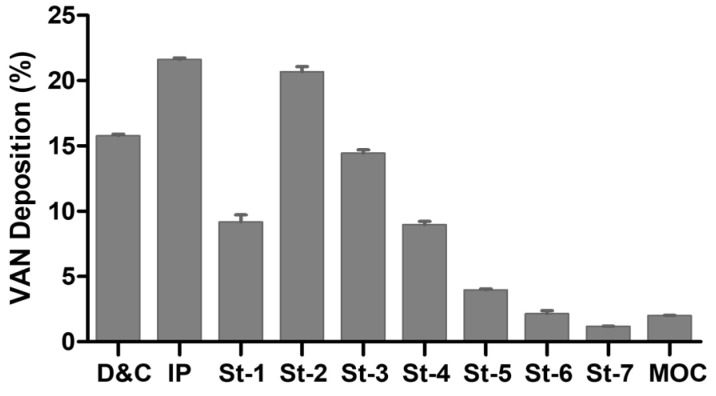
Next-generation impactor deposition of HA-VAN_25_ in the different stages of the equipment: device and capsule (D&C), induction port (IP), stages 1 to 7 (St-1 to St-7) and micro-orifice collector (MOC).

**Figure 6 pharmaceutics-16-00436-f006:**
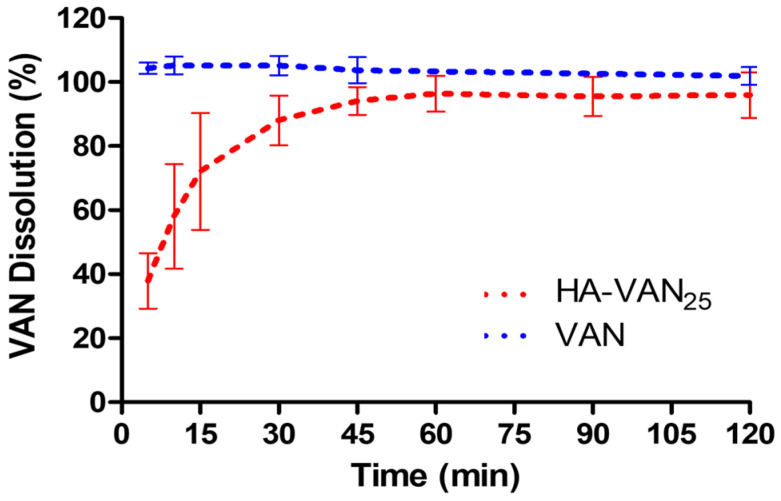
Dissolution profiles of VAN and HA-VAN_25_ in PBS pH 7.4.

**Figure 7 pharmaceutics-16-00436-f007:**
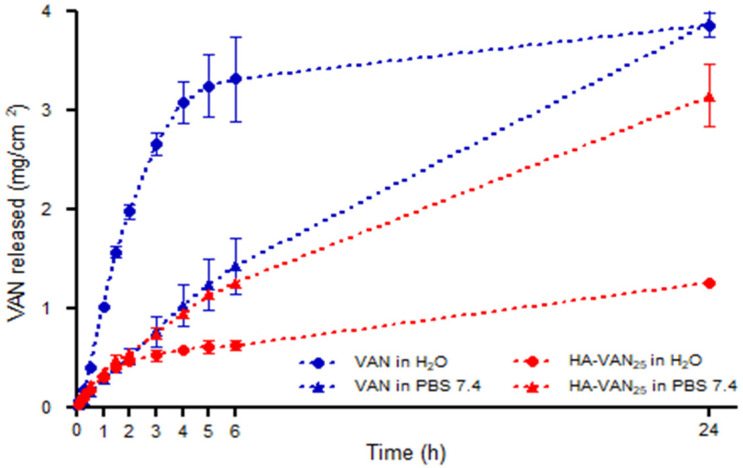
Release profiles of VAN and HA-VAN_25_ towards water and PBS pH 7.4.

**Table 1 pharmaceutics-16-00436-t001:** Aerodynamic parameters of HA-VAN_25_ spray-dried particles obtained by NGI aerosolization.

ED (mg)	EF (%)	FPF (%)	Extra-FPF (%)	MMAD (µm)
7.9 ± 0.4	84.2 ± 0.1	42.9 ± 0.2	25.9 ± 0.1	4.29 ± 0.03

ED: emitted dose, EF: emitted fraction, FPF: fine particle fraction, Extra-FPF: extra-fine particle fraction, MMAD: mass median aerodynamic diameter.

**Table 2 pharmaceutics-16-00436-t002:** Kinetic data obtained from HA-VAN_25_ towards water and PBS pH = 7 using Peppas equation.

Complex	Receptor Medium					
	H20			PBS 7.4		
	*k*	*n*	*R* ^2^	*k*	*n*	*R* ^2^
HA-VAN_25_	0.5	0.60	0.96	0.4	0.67	0.99

**Table 3 pharmaceutics-16-00436-t003:** Halo of inhibition (mm) obtained for HA, VAN and HA-VAN_25_ against methicillin-sensitive *Staphylococcus aureus* (MSSA) and methicillin-resistant *Staphylococcus aureus* (MRSA) ATCC strains.

Samples	Inhibition Halo (mm)		
	ATCC 29213	ATCC 25923	ATCC 43300
HA	8 ± 0.0	8 ± 0.0	8 ± 0.0
VAN	29 ± 0.5 *	31 ± 0.5 *	30 ± 0.5 *
HA-VAN_25_	30 ± 0.5 *	31 ± 0.5 *	31 ± 0.5 *

* *p* < 0.05 respect to HA alone.

**Table 4 pharmaceutics-16-00436-t004:** Minimum inhibitory concentration (MIC) and minimum bactericidal concentration (MBC) values in µg/mL obtained for HA, VAN and HA-VAN_25_ against MSSA and MRSA ATCC strains.

	29213		25923		43300	
Samples	MIC	MBC	MIC	MBC	MIC	MBC
HA	>2500	--	>2500	--	>2500	--
VAN	4.88	4.88	4.88	4.88	4.88	4.88
HA-VAN_25_	4.88	4.88	4.88	4.88	4.88	4.88

## Data Availability

Data are contained within the article.

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
