# Peer review of "Novel Dry Hyaluronic Acid–Vancomycin Complex Powder for Inhalation, Useful in Pulmonary Infections Associated with Cystic Fibrosis"

_pharmaceutics, 2024, doi:10.3390/pharmaceutics16040436_

Round 1

Reviewer 1 Report

Comments and Suggestions for Authors

The manuscript "Novel dry powder of hyaluronic acid-vancomycin complex for inhalation useful on pulmonary infections associated to cystic fibrosis" describes the development and in vitro evaluation of the HA-VAN formulation for pulmonary delivery. Given the overall merit, I suggest the authors substantiate the manuscript with more studies before publication.

1. More discussion and results are warranted in the optimization process of the HA-VAN formulation preparation. e.g., why 25% nebulization of HA was used? 

2. For those in vitro assays, like SEM, dissolution study, cell diffusion study, and antibacterial study, it would be helpful to include a comparison/positive control with a similar VAN pulmonary formulation, if available, to highlight the unique properties of HA-VAN.

3. Those in vitro studies conducted were insufficient to evaluate the in vivo performance of HA-VAN and the local delivery in the infection site compared with IV of VAN, as stated in the aim of the current work. In vivo studies such as PK study assessing local tissue VAN concentration are warranted.

4. More detailed statistical analysis should be included as a standalone paragraph in the method section.

Comments on the Quality of English Language

Please proofread typos and other formatting issues.

Author Response

Thank you very much for taking the time to review this manuscript. Please find attached the detailed responses and the corresponding  corrections in blue color in the re-submitted files.

Reviewer 2 Report

Comments and Suggestions for Authors

The manuscript “Novel dry powder of hyaluronic acid-vancomycin complex for inhalation useful on pulmonary infections associated with cystic fibrosis," which has been submitted to Pharmaceutics by Magi et al., is a highly interesting and well-articulated work. The authors have presented a novel approach to addressing pulmonary infections associated with cystic fibrosis through the use of a hyaluronic acid-vancomycin complex in dry powder form for inhalation.

The paper's strengths lie in its innovative methodology and potential clinical implications. To enhance its suitability for publication, I suggest the following minor improvements:

·         Provide more detailed information regarding the properties of the hyaluronic acid utilized in the study, including its molecular weight, viscosity, etc. Additionally, incorporating a brief commentary on these properties in the introduction would enhance the comprehensiveness of the manuscript.

·         While the dissolution studies conducted were in accordance with the USP, employing PBS as the dissolution medium, it would be beneficial for the authors to explore potential changes in dissolution kinetics in more complex media that mimic real in vivo conditions, such as infected lung mucus. Considering the cationic properties of the drug and the mucoadhesive nature of hyaluronic acid, the authors could provide insights into how these factors may influence dissolution rates in more physiologically relevant environments.

Author Response

(The authors gave the same response as above.)

Reviewer 3 Report

Comments and Suggestions for Authors

This paper is a significant description of the development of a dry powder hyaluronan acid-vancomycin complex " for the treatment of cystic fibrosis and probable for other lung diseases." The paper is a very detailed account of the development of this inhalant and has to be.  For the sake of completion concerning the use of hyaluronic acid, some brief mention of the biological function of hyaluronic acid should be mentioned. Not including such recognition would fail to recognize the various significant beneficial effects that can attend the use of hyaluronic aerosol in cystic fibrosis and would be a significant oversight and detract from the significance of this present paper.

Author Response

(The authors gave the same response as above.)

Reviewer 4 Report

Comments and Suggestions for Authors

The peer-reviewed manuscript “Novel dry powder of hyaluronic acid-vancomycin complex for inhalation useful on pulmonary infections associated to cystic fibrosis” is highly relevant, since the authors propose a new approach to the treatment of pulmonary infections in patients with cystic fibrosis. Inhaled formulations containing vancomycin may be more effective and safer than traditional treatments such as intravenous administration.

The study results indicate that the hyaluronic acid vancomycin complex has suitable aerosol characteristics to reach deep lungs and also has a sustained release of vancomycin, which may improve treatment efficacy.

In summary, the work demonstrates the potential of the new inhaled formulation HA-VAN for the treatment of pulmonary infections in patients with cystic fibrosis, which makes it important and this manuscript will be of interest to readers of Pharmaceutics MDPI.

The article is well designed, illustrated, written in accessible scientific language and, in my opinion, does not require additional edits.

Author Response

Thank you very much for taking the time to review this manuscript. Please find attached the corresponding corrections in blue color in the re-submitted files.

Round 2

Reviewer 1 Report

Comments and Suggestions for Authors

The authors re-submitted the manuscript with all my comments and suggestions addressed. After the second review, I would recommend acceptance of the manuscript.